# Device-aided therapies (DATs) in Parkinson's disease (PD). The DATs-PD GETM Spanish Registry Protocol Study

Diego Santos-García[1,2,3,4]*, Guillermo González-Ortega[5], Pilar Sánchez-Alonso[6], Anna Planas-Ballvé[7], Rocío García-Ramos[8], Iria Cabo[9], Marta Blázquez-Estrada[10], Álvaro Sánchez-Ferro[11,12], DATs-P. D. GETM Spanish Registry Group[¶]

1 CHUAC, Complejo Hospitalario Universitario de A. Coruña, A. Coruña, Spain, 2 Grupo de Investigación en Enfermedad de Parkinson y otros Trastornos del Movimiento, INIBIC (Instituto de Investigación Biomédica de A Coruña), A Coruña, Spain, 3 Hospital San Rafael, A. Coruña, Spain, 4 Fundación Degen, A. Coruña, Spain, 5 Hospital Universitario de Móstoles, Madrid, Spain, 6 Hospital Universitario Puerta de Hierro, Madrid, Spain, 7 Complex Hospitalari Moisès Broggi, Sant Joan Despí, Llobregat, Barcelona, Spain, 8 Hospital Universitario Clínico San Carlos, Madrid, Spain, 9 Complejo Hospitalario Universitario de Pontevedra, Pontevedra, Spain, 10 Hospital Universitario Central de Asturias (HUCA), Oviedo, Spain, 11 Hospital Universitario 12 de Octubre, Madrid, Spain, 12 Department of Medicine, Centro de Investigación Biomédica en Red sobre Enfermedades Neurodegenerativas (CIBERNED), Universidad Complutense, Madrid, Spain

¶ Membership of the DATs-P. D. GETM Spanish Registry Group is listed in the Appendix 1
* diegosangar@yahoo.es

## Abstract

### Background and objective

Device-aided therapies (DATs) are treatments indicated for people with Parkinson´s disease (PwP) experiencing clinical fluctuations that remain suboptimal despite conventional medication. New DATs have recently emerged such as levodopa-entacapone-carbidopa intestinal gel infusion (LECIG) and subcutaneous infusion of foslevodopa/foscarbidopa (fLD/fCD). Understanding the differences between various DATs is essential.

### Patients and Methods

We present here the protocol study of the DATs-PD GETM Spanish Registry. This is a descriptive, observational, prospective, multicenter, open study that is proposed as a clinical registry with progressive inclusion of PwP treated with a DAT in daily clinical practice conditions in more 40 centers from Spain for 10 years. The principal aim is to know the type of DAT that PwP in our country (Spain) receive. Specific objectives are to compare the clinical characteristics of the patients, the effectiveness, safety and tolerability, to identify predictors of a good response and to analyze the response by groups (gender, disease duration, phenotype, etc.). There is a baseline visit (V1; indication of the therapy), start visit (V2; initiation of the therapy) and follow-up visits at 6 months ± 3 months (V3_6M) and after this annually ± 3 months for 10 years (V3_12M, V3_24M, etc.).

**Data availability statement:** No results are shown. This is a Study Protocol. The protocol, statistical analysis plan and deidentified participant data will be available on request (for this project, the registry).

**Funding:** An amount of 25.000 euros has been granted by the "Fundación Professor Novoa Santos" as a result of the "CONVOCATORIA DE AYUDAS PARA LA REALIZACIÓN DE PROYECTOS DE INVESTIGACIÓN PARA GRUPOS EMERGENTES Y ASOCIADOS DEL INIBIC (2023/2024)". The import will be used for data monitoring and other actions required in the context of the project.

**Competing interests:** The authors have declared that no competing interests exist.

**Abbreviations:** ADL, activities of daily living; CSAI, continuous subcutaneous apomorphine infusion; DATs, device-aided therapies; DBS, deep brain stimulation; fLD/fCD, foslevodopa-foscarbidopa; H&Y, Hoehn & Yahr; LCIG, levodopa-carbidopa intestinal gel infusion; LECIG, levodopa-entacapone-carbidopa intestinal gel infusion; LEED, levodopa equivalent daily dose; MMSE, Mini-Mental State Examination; MoCA, Montreal Cognitive Assessment; PD, Parkinson´s disease; PD-CRS, Parkinson´s Disease Cognitive Rating Scale; UPDRS, Unified Parkinson´s Disease Rating Scale.

## Results

The registry is on going. The first patient was included on April 10, 2024. Patient recruitment and follow-up will be conducted until 31/DEC/2033. It is estimated that the registry will include a minimum of 3,000 patients.

## Conclusion

The present study will help improve the care of PD patients treated with a DAT.

## Introduction

Parkinson´s disease (PD) is the second most common neurodegenerative disease after Alzheimer's disease. It is characterized by a deficit of dopamine in the striatum and other brain areas, but also of other neurotransmitters such as noradrenaline, acetylcholine or serotonin, which would explain the appearance of motor and non-motor symptoms characteristic of the disease [1]. The diagnosis is made by applying well-defined criteria [2,3] that are based fundamentally on the existence of parkinsonism, the absence of atypical data that suggest an alternative diagnosis (pharmacological, atypical parkinsonism, etc.), and data in favor that suggest PD itself. Among the latter, a determining aspect is the good response to dopaminergic medication, especially levodopa [4]. Its administration compensates for the deficit of dopaminergic stimulation at the level of the postsynaptic receptors, which causes an improvement in symptoms such as tremor, rigidity or bradykinesia, among others. In fact, the absence of a response would go against the diagnosis of PD.

Although the response to dopaminergic medication may be optimal during the initial years, people with PD (PwP) often develop motor and non-motor complications (e.g., motor fluctuations, non-motor fluctuations, dyskinesias, etc.) that affect their quality of life and autonomy [5]. Thus, PwP may only perceive improvement at certain times of the day alternating with disabling OFF episodes where the symptoms reemerge and the control with conventional medication is insufficient [6]. Some PwP may benefit from treatment with a second-line therapy that is an alternative and/or complement to conventional medication that is not sufficient [7]. The proper selection of the candidates is key, and there are different tools for their correct identification [8]. These therapies are commonly known as device-aided therapies (DATs) and although they are more expensive and complex than conventional medication, they have been shown to reduce OFF time, improve non-motor symptoms and the quality of life of PwP [9]. For many years the primary DATs available in numerous countries have been the deep brain stimulation (DBS), continuous subcutaneous apomorphine infusion (CSAI) and levodopa-carbidopa intestinal gel infusion (LCIG) [10]. However, new treatments have recently emerged, such as levodopa-entacapone-carbidopa intestinal gel infusion (LECIG) and subcutaneous infusion of foslevodopa-foscarbidopa (fLD/fCD) [11,12]. Particularly noteworthy is the recent availability of subcutaneous fLD/fCD infusion, to the point that many recently published treatment algorithms are outdated and the possibility of considering the subcutaneous route as the first alternative to the enteral route is being discussed, as it is less invasive when considering infusion therapy [13,14].

Therefore, a completely new scenario is opening up in the treatment of PwP with a DAT and it is essential to know the frequency of the prescriptions, the characteristics of the treated individuals as well as their long-term evolution in relation to clinical changes and complications. This is the reason why, from the Movement Disorders Study Group (Grupo de Estudio de Trastornos de Movimiento [GETM]) of the Spanish Neurological Society (Sociedad

Española de Neurología [SEN]), we have launched a prospective registry of PwP treated with a DAT in our country (Spain). The aim of this article is to describe this registry that we have named DATs-PD GETM Spanish Registry.

## Methods/design

**Title of the Project:** Device-aided therapies in Parkinson´s disease GETM Spanish Registry (DATs-PD GETM Spanish Registry).

**Type of study:** Descriptive, observational, prospective, multicenter, open study that is proposed as a clinical registry with progressive inclusion of patients with PwP treated with a DAT in daily clinical practice conditions.

**Promoter and principal coordinator of the project:** Diego Santos García, MD, PhD.

**Coordinating institutions**: "Fundación Degen", "Grupo de Estudio de Trastornos del Movimiento (GETM)" and "Sociedad Española de Neurología (SEN)".

**Participating centers in the project:** More than 40 centers from Spain with neurology teams with experience in the management of PD (**Appendix 1**), including the use of DATs.

**Population:** PwP treated with a DAT in Spain from year 2024. The following therapies are included as DATs: DBS; CSAI; LCIG; LECIG; and fLD/fCD subcutaneous infusion. In addition, PwP who are treated with focused ultrasound-guided thermal ablation will also be included. Specifically, the eligibility criteria are: 1) diagnosis of PD according to the MDS criteria [3]; 2) start of treatment with a DAT from January 1, 2024; 3) the patient's desire to participate on a completely voluntary basis; 4) signing of an informed consent.

**Justification of the project:** 1) there are many DATs currently used to treat PwP; 2) some of them have been introduced very recently and we have no experience; 3) in this context, we must know about the indication preferences in Spanish centers that treat PwP; 4) we must know the characteristics of the individuals treated and the differences between therapies; 4) we must know the effectiveness of DATs and to compare them; 5) we must know the safety and tolerability of DATs and to compare them; 6) it is necessary to define different response profiles or predictors of better or worse outcome; 7) this is an excellent opportunity to be able to develop a registry at this time that collects a lot of information over the long term; 8) It could be a starting point to which centers from other countries can join and develop a broader registry (e.g., European, etc.).

### Objectives

**Principal objective**: to know the type of DAT that PwP in our country receive, treated by expert neurologists in daily clinical practice conditions.

**Specific objectives**: 1) to analyze the sociodemographic and clinical characteristics of PwP treated with a DAT, comparing the different treatments (DBS vs subcutaneous vs enteral treatment); 2) to analyze the effectiveness of different DATs, comparing the different treatments (DBS vs subcutaneous vs enteral treatment); 3) to analyze the safety and tolerability of different DATs, comparing the different treatments; 4) specifically, to analyze the changes experienced by PwP treated with DATs in the perception of their quality of life and autonomy for carrying out daily activities and to compare the different treatments; 5) specifically, to compare the rate of maintenance of the therapy between the different DAT groups; 6) specifically, to compare the dropout rate for each of the therapies and find out the underlying reasons; 7) to find out the reasons for changing from one therapy to another or adding a second DAT to a previous one being received; 8) to analyze the complications in relation to the different DATs and compare them; 9) to define predictors of a favorable response to different DATs; 10) to

compare the differences by gender (male vs. female); 11) others that may be raised based on the data collected (e.g., differences by age, disease duration etc.).

## Visits

The registry includes 3 types of visits: 1) baseline visit (V1), which is when the DAT is decided by the neurologist; 2) start visit (V3), which is when the DAT is initiated by the patient; 3) follow-up visit (V3), with the patient receiving the DAT. The first follow-up visit will be carried out at 6 months +/- 3 months (V3_6M) and then at 1 year +/-3 months and subsequently annually +/- 3 months: 1 year (V3_12M); 2 years (V3_24M); 3 years (V3_36M); 4 years (V3_48M); 5 years (V3_60M); 6 years (V3_72M); 7 years (V3_84M); 8 years (V3_96M); 9 years (V3_108M); 10 years (V3_120M) (**Fig 1**).

Patient inclusion in the registry will span a minimum of 5 years, with the option to extend to 10 years as per the approved protocol. New PwP will be included over a 10-year period, while each participant's included will be follow-up during this time. The data collection period ranges from January 1, 2024 to December 31, 2033. If a person with PD switches to another DAT or a second DAT is added, the information will be collected. In addition, if a subject drops out of the DAT and does not receive another DAT, data collection will continue within the registry to allow for a comparative arm of untreated patients after a therapeutic failure (something rarely discussed in the literature). Although the registry is prospective, it

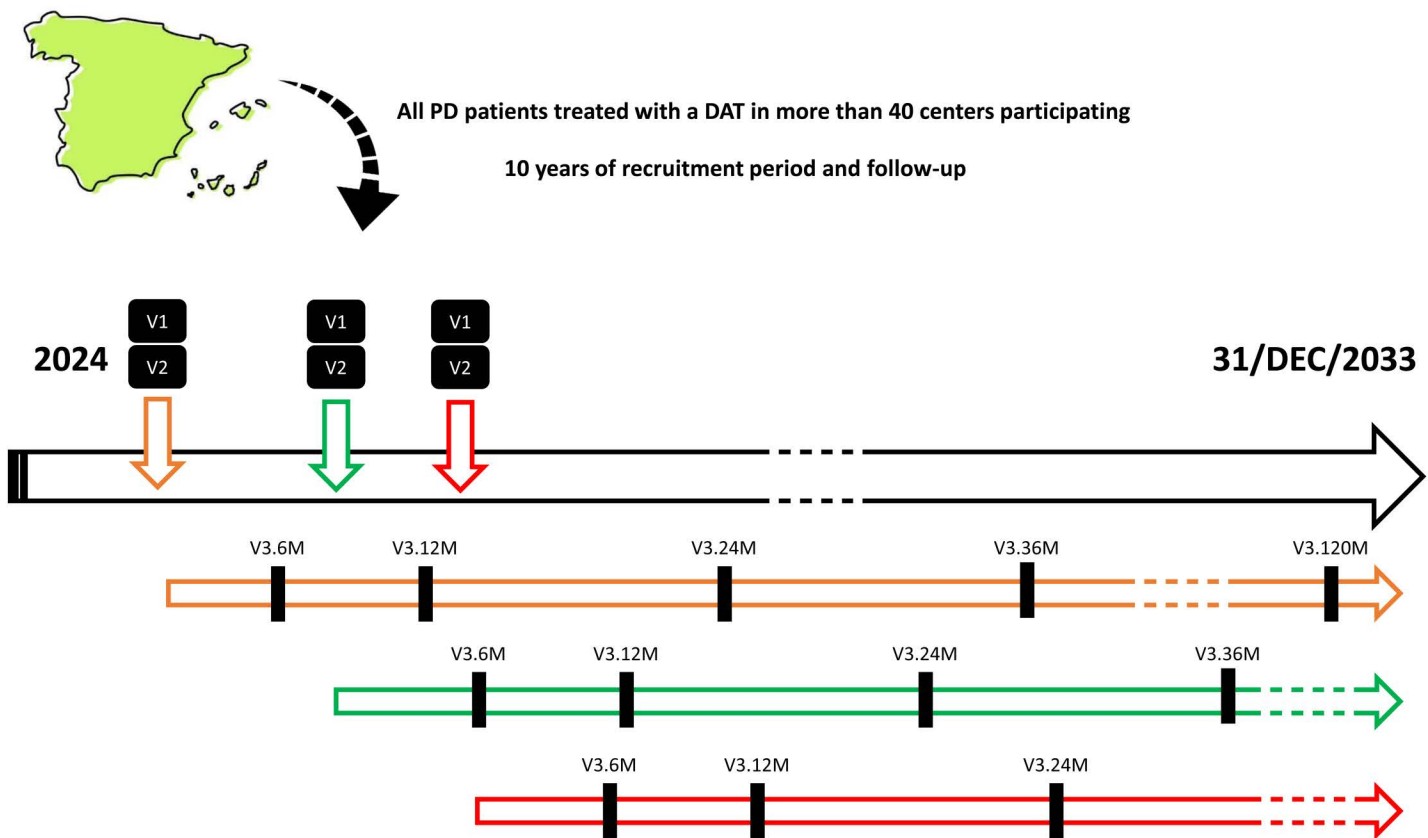

**Fig 1. All patients treated with a DAT from 2024 to 31/DEC/2033 in more than 40 participating centers from Spain are included, all of them evaluated by neurologists who are experts in PD.** Each color refers to a different patient. There is a baseline visit (V1; indication of the therapy), start visit (V2; initiation of the therapy) and follow-up visits at 6 months (V3_6M) and after this annually (V3_12M, V3_24M, etc.). The window for V2 and V3 is ± 3 months.

is planned to include PwP treated in 2024 whose indication for DAT was before this year, as long as the data was properly collected in the medical record. An example is DBS, given that the waiting list from indication to intervention is long in some centers.

## Assessments

The following information will be collected at each visit:

1) Baseline visit (V1): sociodemographic data; data about PD (age onset, disease duration, motor phenotype, etc.); comorbidities; treatments; main reason for therapy indication; levodopa equivalent daily dose (LEDD) [15]; motor symptoms; non-motor symptoms including cognition; the Mini-Mental State Examination (MMSE) [16], Montreal Cognitive Assessment (MoCA) [17] or Parkinson´s Disease Cognitive Rating Scale (PD-CRS) [18] may be used at the discretion of the neurologist in daily clinical practice); Unified Parkinson´s Disease Rating Scale (UPDRS-III) [19] during the OFF state; UPDRS-III during the ON state; Hoehn & Yahr (H&Y) [20] stage in OFF; H&Y stage in ON; MNCD (Motor/Non-motor/Cognition/Dependency) classification (classification; score; stage) [21]; Parkinson's Disease Questionnaire (PDQ-39) [22]; European Health Interview Survey-Quality of Life 8-item index (EUROHIS-QOL 8-item index) [23]; Schwab and England Activities of Daily Living Scale (ADLS) [24] during the OFF and during the ON state; serum levels of folate, B vitamins (B1, B6, B12) and homocysteine.

2) Start visit (V2): data about the DAT and treatment settings including LEDD.

3) Follow-up visit (V3): data about the DAT and treatment settings including LEDD; comorbidities; treatments; LEDD; motor symptoms; non-motor symptoms; UPDRS-III during the ON state; H&Y stage in ON; MNCD classification (classification; score; stage); Parkinson's Disease Questionnaire (PDQ-39); EUROHIS-QOL 8-item index; ADLS during the OFF and during the ON state; serum levels of folate, B vitamins (B1, B6, B12) and homocysteine; complications about the DAT and/or other relevant complications.

All information will be collected by expert neurologists in a daily clinical practice setting. A total of 16 and 26 questions collect information on motor symptoms/complications and non-motor symptoms, respectively, which are categorized as absent, mild, moderate, severe, or very severe. The score on the MDS-UPDRS-III scale [25] will be an added option that will be collected in those centers that use it (permission by the MDS was obtained). **Table 1** summarizes the information on the variables that will be collected during the different visits.

## Data collection and statistical analysis

Data will be collected using REDCap. This is a secure web application for building and managing online surveys and databases. REDCap has been used to date (27/NOV/2024) in 159 countries by more than 3.4 million users. Data collected will be transferred to a statistical package for subsequent analysis. The promoter team of the project will be responsible for study monitoring. The estimate, with the participation of more than 40 centers and a recruitment period of 10 years, is to be able to reach a minimum sample size of 3,000 patients. The pertinent analysis (descriptive, missing data analysis, normality assumptions, univariate, binary logistic regression, multiple linear regression, etc.) will be performed based on the type of objective. In addition, given de complexity of potential analysis including a diversity of variables from different origin and measurement properties, advanced statistical methodology (data mining, machine learning or other artificial intelligence techniques, etc.) could be considered when applicable.

## Standard protocol approvals, registrations, and patient consents

The project will be conducted in accordance with the ICH Good Clinical Practice version 6 Revision 2 standard, the fundamental ethical principles established in the Declaration of Helsinki and the Oviedo Convention, as well as the Spanish legal requirements for biomedical research (Biomedical Research Law 14/2007). The Project has been approved on 02/APR/2024 by the IRB "Comité de Ética de la Investigación Clínica de Galicia from Spain" with code number 2024/109. Written informed consents from all participants in this study will be obtained.

## Study timetable

1) Pre-start-up procedures: until April 2024.

2) First patient included: 10/APR/2024.

3) First analysis to present: 2025 – Q1 (data about PD patients treated in 2024).

4) Proposed specific objectives and others: from 2025 onwards.

**Table 1. Variables collected in the DAT-PD Spanish Registry according to the type of visit.**

| V1 | V2 | V3* |
|---|---|---|
| Sociodemographic | Information about DAT | Information about DAT |
| PD history | Concomitant treatments | Concomitant treatments |
| Comorbidities | LEDD | LEDD |
| Concomitant treatments | | PD symptoms |
| LEDD | | Motor symptoms |
| Main reason for DAT | | Non-motor |
| PD symptoms | | symptoms** |
| Motor symptoms | | Motor scales |
| Non-motor | | UPDRS-III-OFF |
| symptoms** | | UPDRS-III ON |
| Motor scales | | H&Y-OFF |
| UPDRS-III-OFF | | H&Y-ON |
| UPDRS-III ON | | Quality of life |
| H&Y-OFF | | PDQ-39 |
| H&Y-ON | | EUROHIS-QOL8 |
| Quality of life | | Functional dependency |
| PDQ-39 | | ADLS-OFF |
| EUROHIS-QOL8 | | ADLS-ON |
| Functional dependency | | Serum biomarkers |
| ADLS-OFF | | Folate |
| ADLS-ON | | Vitamin B1 |
| Serum biomarkers | | Vitamin B6 |
| Folate | | Vitamin B12 |
| Vitamin B1 | | Homocysteine |
| Vitamin B6 | | Complications |
| Vitamin B12 | | Related with DAT |
| Homocysteine | | Unrelated with DAT |

*Information collected for all these visits (with a window of +/- 3 months): 1 year (V3_12M); 2 years (V3_24M); 3 years (V3_36M); 4 years (V3_48M); 5 years (V3_60M); 6 years (V3_72M); 7 years (V3_84M); 8 years (V3_96M); 9 years (V3_108M); 10 years (V3_120M).

**Including cognition; the Mini-Mental State Examination (MMSE), Montreal Cognitive Assessment (MoCA) or Parkinson´s Disease Cognitive Rating Scale (PD-CRS) may be used at the discretion of the neurologist in daily clinical practice.

ADLS, Schwab & England Activities of Daily Living Scale; DAT, device-aided therapy; LEDD, levodopa equivalent daily dose [15]; PD, Parkinson´s disease; PDQ-39, the 39-item Parkinson's disease Questionnaire; UPDRS, Unified Parkinson´s Disease Rating Scale.

## Strengths and limitations of the project

The main strength of the project is the possibility of collecting systematic information on patients treated with DATs over a long follow up period (10 years). Further, there is great interest in the fact that some of these therapies are novel and the management of advanced PD patients is changing because of this and could hence be evaluated. Also, the methodology with prospective follow-up and the collection of well-defined variables (UPDRS, H&Y, PDQ-39, ADLS, etc.) in a clinical practice setting will allow generalizing the findings to other environments. To this end, we also consider in the future increasing the number of centers in Spain, and even the chance of extrapolating the registry to other countries harmonizing the data collected. Other future possibilities are implementing new technologies to monitor the disease and analyze the changes after starting with the DAT and their correlation with the data collected from the registry.

On the other hand and as limitations, unlike a study with a clearly defined protocol in which multiple scales are administered [26], this project is a registry, and limitations in the quantity and quality of the data collected are possible. For example, the number and description of complications may be underestimated. At this moment the project is exciting but there is some residual risk that it will not work properly due to the workload of researchers.

## Discussion

According to the PARADISE study, a non-interventional, cross-sectional, multicenter, national study conducted in the hospital setting and published in 2021, up to 38.2% of people with PD in Spain have advanced PD [27]. The key aspect of identifying advanced PD is to differentiate which PwP could be candidates to receive a DAT. However, only 15.2% of PwP from the advanced PD group from the PARADISE study were receiving some form of therapy for advanced stages of the disease (i.e., DBS, CSAI, LCIG). The most frequent reasons why advanced PwP were not on a DAT were "to be clinically stable" and "option not yet considered". Similar results were observed in the PROSPECT study [28], in which only 20.9% of PwP (adults with levodopa-responsive PD and inadequately controlled motor symptoms with ≥ 2.5-hours/day "Off" time, despite trials of available oral/transdermal/sublingual/inhalable medication) initiated a DAT despite the fact that, as expected, DAT vs best medical therapy improved motor symptoms, non-motor symptoms, sleep, quality of life, independence for ADL and caregiver burden. For many years, the options for being treated with a DAT meant choosing between three alternatives: DBS, CSAI, LCIG. However, the scenario is new with the current availability of LECIG and subcutaneous fLD/fCD. In particular, it is of maximum interest to know whether the arrival of fLD/fCD implies a reduction in the indication of other treatments and what are the characteristics of the treated patients as well as their complications in comparison with other DATs, especially enteral therapies [29]. This registry aims to answer this question and many others explained in the objectives. We believe that the information collected in the DATs-PD GETM Spanish Registry will be of great interest in helping to improve the management of PwP treated with a DAT.

In summary, we present here the protocol study of the DATs-PD GETM Spanish Registry, a descriptive, observational, prospective, multicenter, open study that is proposed as a clinical registry with progressive inclusion of PwP treated with a DAT in daily clinical practice conditions in more 40 centers from Spain for 10 years. The present study will help improve the care of PwP treated with a DAT.

## Acknowledgements

We would like to thank all patients who collaborated in this study. Many thanks also to Fundación Española de Ayuda a la Investigación en Enfermedades Neurodegenerativas y/o de Origen Genético (https://fundaciondegen.org/), Grupo de Estudio de Trastornos del

Movimiento (GETM), Sociedad Española de Neurología (SEN) and Fundación Profesor Novoa Santos. Also, thanks to Steve Winslow from Scholar Memory Academic Writing Consultant for adding improvements to the English style of the manuscript.

## Author contributions

**Conceptualization:** Diego Santos Garcia.

**Investigation:** Diego Santos Garcia, Guillermo González-Ortega, Pilar Sánchez-Alonso, Ana Planas-Ballvé, Rocío García-Ramos, Iria Cabo, Marta Blazquez-Estrada, Álvaro Sánchez-Ferro, STUDY GROUP DATs-PD GETM Spanish Registry.

**Methodology:** Diego Santos Garcia, Guillermo González-Ortega.

**Project administration:** Diego Santos Garcia, Guillermo González-Ortega, Álvaro Sánchez-Ferro.

**Resources:** Diego Santos Garcia, Álvaro Sánchez-Ferro.

**Supervision:** Diego Santos Garcia.

**Validation:** Diego Santos Garcia.

**Visualization:** Diego Santos Garcia.

**Writing – original draft:** Diego Santos Garcia.

**Writing – review & editing:** Guillermo González-Ortega, Pilar Sánchez-Alonso, Rocío García-Ramos, Iria Cabo, Marta Blazquez-Estrada, Álvaro Sánchez-Ferro, STUDY GROUP DATs-PD GETM Spanish Registry.

## Appendix 1

Adarmes Gómez AD, Alonso Modino D, Álvarez Sauco M, Aneiros Díaz A, Ávila Rivera A, Baviera-Muñoz R, Belmonte S, Blázquez Estrada M, Caballero Sánchez L, Caballol Pons N, Cabo I, Campins Romeu M, Campolongo A, Cantarero Duque S, Carrillo García F, Casanova Mollá J, Casas Peña E, Castaño García B, Castellano Guerrero AM, Castrillo Sanz A, Cerdán Santacruz DM, Clavero Ibarra P, Cores Bartolomé C, Cots Foraster A, Cubo Delgado E, Delgado Ballestero T, Erdocia Goñi A, Escalante Arroyo S, Escamilla Sevilla F, Espinosa Rosso R, Fanjul Arbos S, Feliz Feliz C, Fernández-Pajarín G, Fernández Revuelta A, Fernández Rodríguez B, Fernández Valle T, Freire Álvarez E, Gamo González E, García Fernández C, García Herruzo A, García Ramos García R, García Ruíz Espiga P, Garrote Espina L, Gil Villar MP, Gómez Esteban JC, Gómez López de San Román C, Gómez Mayordomo V, Gómez Rapela C, González MV, González Ardura J, González Hernández A, González-Ortega G, Guerra Hiraldo JD, Gutiérrez García J, Hernández Vara J, Jesús Maestre S, Kulisevsky Bojarski J, Legarda Ramírez I, López Ariztegui N, López Blanco R, López Dominguez D, López Manzanares L, López Veloso AC, López Valdés E, Lorenzo Barreto P, Lorenzo Brito JM, Lozano D, Macías García D, Madrid Navarro CJ, Martí Andrés G, Martí Martínez S, Martín García R, Martínez Castrillo JC, Martínez-Torres I, Mata Álvarez Santullano M, Mauri Fabrega L, Méndez Guerrero A, Mendoza Rodríguez A, Mir P, Mondragón Rezola E, Monterde Ortega A, Morales Casado MI, Morata-Martínez C, Muñoz Delgado L, Muñoz Ruíz T, Muro I, Novo Ponte S, Ojeda Lepe E, Olivares Romero J, Pagonabarraga J, Pareés Moreno I, Pascual Sedano B, Paz González JM, Peral Quirós A, Pérez Calvo C, Pérez Rangel D, Perona Moratalla A, Planas-Ballvé A, Prendes Fernández P, Quibus Requena L, Rábano Suárez P, Rashid López R, Rebollo Lavado B, Rivero de Aguilar Pensado A, Rojas Pérez E, Ribacoba Díaz C, Romero Fábrega JC, Ruiz López M, Ruíz Martínez J, Samaniego Vinueza LB, San Eufrasio Martínez M, Sánchez Alonso P, Sánchez

Ferro A, Sánchez Rodríguez A, Sancho Saldana A, Santos-García D, Sastre Bataller I, Solano Vila B, Solleiro Vidal A, Suárez San Martín E, Tabar Comellas G, Tijero Merino B, Triguero Cueva L, Valero García MF, Valldeoriola Serra F, Vela L, Vinagre Aragón A, Vivas Villacampa L, Vives Pastor B, Yáñez Baña R.

| First Name | Last Name | Centre |
|---|---|---|
| Diego | Santos García | Complejo Hospitalario Universitario de A Coruña |
| Jose Manuel | Paz González | Complejo Hospitalario Universitario de A Coruña |
| Carlos | Cores Bartolomé | Complejo Hospitalario Universitario de A Coruña |
| Lucia Belen | Samaniego Vinueza | Complejo Hospitalario Universitario de A Coruña |
| Ángela | Solleiro Vidal*** | Complejo Hospitalario Universitario de A Coruña |
| María | Álvarez Sauco | Hospital General Universitario de Elche |
| Eric | Freire Álvarez | Hospital General Universitario de Elche |
| Juan Carlos | Martínez Castrillo | Hospital Universitario Ramón y Cajal |
| Isabel | Pareés Moreno | Hospital Universitario Ramón y Cajal |
| Samira | Fanjul Arbos | Hospital Universitario Ramón y Cajal |
| Ana Belén | Perona Moratalla | Complejo Hospitalario Universitario de Albacete |
| Inés | Legarda Ramírez | Hospital Universitario Son Espases |
| Bàrbara | Vives Pastor | Hospital Universitario Son Espases |
| María Fuensanta | Valero García | Hospital Universitario Son Espases |
| Esther | Cubo Delgado | Hospital Universitario de Burgos |
| Nuria | López Ariztegui | Hospital Universitario de Toledo |
| Maria Isabel | Morales Casado | Hospital Universitario de Toledo |
| Guillermo | Tabar Comellas | Hospital Universitario de Toledo |
| Déborah | Alonso Modino | Hospital Universitario de la Candelaria |
| Jesús Norelis | Lorenzo Brito | Hospital Universitario de la Candelaria |
| Esther | Rojas Pérez | Hospital Universitario de la Candelaria |
| Iria | Cabo | Complejo Hospitalario Universitario de Pontevedra |
| Alejandro | Rivero de Aguilar Pensado | Complejo Hospitalario Universitario de Pontevedra |
| Jorge | Hernández Vara | Hospital Universitario Vall d´Hebron |
| Maria Victoria | González | Hospital Universitario Vall d´Hebron |
| Sara | Belmonte** | Hospital Universitario Vall d´Hebron |
| Juan Carlos | Romero Fábrega | Hospital Universitario Virgen de las Nieves |
| Francisco | Escamilla Sevilla | Hospital Universitario Virgen de las Nieves |
| Lucía | Triguero Cueva | Hospital Universitario Virgen de las Nieves |
| Carlos Javier | Madrid Navarro | Hospital Universitario Virgen de las Nieves |
| Rosa | Yáñez Baña | Complejo Hospitalario Universitario de Ourense |
| Asunción | Ávila Rivera | Complex Hospitalari Moisès Broggi |
| Núria | Caballol Pons | Complex Hospitalari Moisès Broggi |
| Anna | Planas-Ballvé | Complex Hospitalari Moisès Broggi |
| Alejandro | Peral Quirós | Complex Hospitalari Moisès Broggi |
| Dolors | Lozano | Complex Hospitalari Moisès Broggi |
| Álvaro | Sánchez Ferro | Hospital 12 de Octubre |
| Pablo | Rábano Suárez | Hospital 12 de Octubre |
| Antonio | Méndez Guerrero | Hospital 12 de Octubre |
| Daniel | Pérez Rangel | Hospital 12 de Octubre |
| Frances | Valldeoriola Serra | Hospital Clínic |
| Rocío | García Ramos | Hospital Clínico Universitario San Carlos |
| Ana | Fernández Revuelta | Hospital Clínico Universitario San Carlos |
| Eva | López Valdés | Hospital Clínico Universitario San Carlos |

| First Name | Last Name | Centre |
| --- | --- | --- |
| Carmen | Ribacoba Díaz | Hospital Clínico Universitario San Carlos |
| Irene | Martínez-Torres | Hospital Universitario la Fe |
| Carlos | Morata-Martínez | Hospital Universitario la Fe |
| Raquel | Baviera-Muñoz | Hospital Universitario la Fe |
| Marina | Campins Romeu | Hospital Universitario la Fe |
| Isabel | Sastre Bataller | Hospital Universitario la Fe |
| Pilar | Sánchez Alonso | Hospital Puerta de Hierro |
| Sabela | Novo Ponte | Hospital Puerta de Hierro |
| Elisa | Gamo González | Hospital Puerta de Hierro |
| Raquel | Martín García | Hospital Puerta de Hierro |
| Pablo | Mir | Hospital Universitario Virgen del Rocío |
| Laura | Muñoz Delgado | Hospital Universitario Virgen del Rocío |
| Astrid Daniela | Adarmes Gómez | Hospital Universitario Virgen del Rocío |
| Elena | Ojeda Lepe | Hospital Universitario Virgen del Rocío |
| Silvia | Jesús Maestre | Hospital Universitario Virgen del Rocío |
| Daniel | Macías García | Hospital Universitario Virgen del Rocío |
| Fátima | Carrillo García | Hospital Universitario Virgen del Rocío |
| Ana María | Castellano Guerrero*** | Hospital Universitario Virgen del Rocío |
| Manuela | San Eufrasio Martínez*** | Hospital Universitario Virgen del Rocío |
| Cristina | Pérez Calvo*** | Hospital Universitario Virgen del Rocío |
| Andrés | García Herruzo | Hospital Universitario Virgen del Rocío |
| Cristina | Gómez Rapela***** | Hospital Universitario Virgen del Rocío |
| Lorena | Garrote Espina***** | Hospital Universitario Virgen del Rocío |
| Marta | Blázquez Estrada | Hospital Universitario Central de Asturias |
| Esther | Suárez San Martín | Hospital Universitario Central de Asturias |
| Ciara | García Fernández | Hospital Universitario Central de Asturias |
| Patricia | Prendes Fernández**** | Hospital Universitario Central de Asturias |
| Pedro | García Ruíz Espiga | Hospital Fundación Jiménez Díaz |
| Cici | Feliz Feliz | Hospital Fundación Jiménez Díaz |
| Raúl | Espinosa Rosso | Hospital Universitario de Jerez |
| Lara | Mauri Fabrega | Hospital Universitario de Jerez |
| Jaime | Kulisevsky Bojarski | Hospital Sant Pau de Barcelona |
| Berta | Pascual Sedano | Hospital Sant Pau de Barcelona |
| Javier | Pagonabarraga | Hospital Sant Pau de Barcelona |
| Antonia | Campolongo** | Hospital Sant Pau de Barcelona |
| Marina | Mata Álvarez-Santullano | Hospital Universitario Infanta Sofía |
| Juan Carlos | Gómez Esteban | Hospital de Cruces |
| Tamara | Fernández Valle | Hospital de Cruces |
| Marta | Ruiz López | Hospital de Cruces |
| Beatriz | Tijero Merino | Hospital de Cruces |
| Lydia | López Manzanares | Hospital Universitario La Princesa |
| Inés | Muro | Hospital Universitario La Princesa |
| Elena | Casas Peña | Hospital Universitario La Princesa |
| Pablo | Lorenzo Barreto | Hospital Universitario La Princesa |
| Maria Pilar | Gil Villar | Hospital Universitari Arnau de Vilanova |
| Agustín | Sancho Saldana | Hospital Universitari Arnau de Vilanova |
| Laura | Quibus Requena | Hospital Universitari Arnau de Vilanova |
| Jesús | Olivares Romero | Hospital Universitario de Torrecárdenas |

| First Name | Last Name | Centre |
|---|---|---|
| Belén | Rebollo Lavado | Hospital Universitario de Torrecárdenas |
| Lucía | Triguero Cueva* | Hospital Universitario de Torrecárdenas |
| Víctor | Gómez Mayordomo | Hospitales Vithas |
| Berta | Solano Vila | Hospital Josep Trueta de Girona y Hospital Santa Caterina de Salt |
| Anna | Cots Foraster | Hospital Josep Trueta de Girona y Hospital Santa Caterina de Salt |
| Daniel | López Dominguez | Hospital Josep Trueta de Girona y Hospital Santa Caterina de Salt |
| Lilian | Vivas Villacampa | Hospital Josep Trueta de Girona y Hospital Santa Caterina de Salt |
| Jessica | González Ardura | Hospital de Cabueñes |
| Belén | Castaño García | Hospital de Cabueñes |
| Antonio | Sánchez Rodríguez | Hospital de Cabueñes |
| Sonia | Escalante Arroyo | Hospital Virgen de la Cinta |
| Tania | Delgado Ballestero | Parc Taulí |
| Débora María | Cerdán Santacruz | Hospital General de Segovia |
| Amelia | Mendoza Rodríguez | Hospital General de Segovia |
| Ana | Castrillo Sanz | Hospital General de Segovia |
| Lorena | Caballero Sánchez | Hospital General de Segovia |
| Claudia | Gómez López de San Román | Hospital General de Segovia |
| Gustavo | Fernández-Pajarín | Complejo Hospitalario Universitario de Santiago de Compostela |
| Jordi | Casanova Mollá | Hospital Universitario Sant Joan de Reus |
| Ángela | Monterde Ortega | Hospital Universitario Joan XXIII |
| Susana | Cantarero Duque | Hospital Universitario de Móstoles |
| Guillermo | González-Ortega | Hospital Universitario de Móstoles |
| Beatriz | Fernández Rodríguez | Hospital Regional Universitario de Málaga |
| Teresa | Muñoz Ruíz | Hospital Regional Universitario de Málaga |
| Gloria | Martí Andrés | Hospital Universitario de Navarra, Pamplona, Navarra |
| Pedro | Clavero Ibarra | Hospital Universitario de Navarra, Pamplona, Navarra |
| Amaia | Erdocia Goñi | Hospital Universitario de Navarra, Pamplona, Navarra |
| Ana Carolina | López Veloso | Hospital Universitario Doctor Negrín, Las Palmas de Gran Canaria |
| Ayoze Nauzet | González Hernández | Hospital Universitario Doctor Negrín, Las Palmas de Gran Canaria |
| Juan Diego | Guerra Hiraldo | Hospital Universitario Doctor Negrín, Las Palmas de Gran Canaria |
| Raúl | Rashid López | Hospital Puerta del Mar, Cádiz |
| Raúl | Espinosa Rosso | Hospital Puerta del Mar, Cádiz |
| Silvia | Martí Martínez | Hospital General Universitario Alicante |
| Ángel | Aneiros Díaz | Complejo Hospitalario Universitario de Ferrol |
| Lydia | Vela | Hospital Fundación de Alcorcón |
| Javier | Ruíz Martínez | Hospital Universitario Donostia |
| Elisabet | Mondragón Rezola | Hospital Universitario Donostia |
| Ana | Vinagre Aragón | Hospital Universitario Donostia |
| Javier | Gutiérrez García | Complejo Hospitalario Clínico San Cecilio, Granada |
| Roberto | López Blanco | Hospital Universitario Severo Ochoa, Leganés, Madrid |

The researchers are listed by center in chronological order according to the date they confirmed their participation in the project.

*This investigator works at 2 different centers.

All investigators are neurologists except those ones with the symbol:

**SN (specialized nurse);

***graduate in Biology (she will be responsible for remote data monitoring);

****neuropsychologist;

*****, study coordinator.

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
