## [Decision Letter · Decision Letter 0]

21 Jan 2025

PONE-D-24-54866Device-aided therapies (DATs) in Parkinson´s disease (PD). The DATs-PD GETM Spanish Registry.PLOS ONE

Dear Dr. Santos Garcia,

Thank you for submitting your manuscript to PLOS ONE. After careful consideration, we feel that it has merit but does not fully meet PLOS ONE’s publication criteria as it currently stands. Therefore, we invite you to submit a revised version of the manuscript that addresses the points raised during the review process.

We look forward to receiving your revised manuscript.

Kind regards,

Barry Kweh

Academic Editor

PLOS ONE

Journal requirements: When submitting your revision, we need you to address these additional requirements. 1. Please ensure that your manuscript meets PLOS ONE's style requirements, including those for file naming. The PLOS ONE style templates can be found at https://journals.plos.org/plosone/s/file?id=wjVg/PLOSOne_formatting_sample_main_body.pdf and https://journals.plos.org/plosone/s/file?id=ba62/PLOSOne_formatting_sample_title_authors_affiliations.pdf. 2. Please amend either the title on the online submission form (via Edit Submission) or the title in the manuscript so that they are identical. 3. We suggest you thoroughly copyedit your manuscript for language usage, spelling, and grammar. If you do not know anyone who can help you do this, you may wish to consider employing a professional scientific editing service.  The American Journal Experts (AJE) (https://www.aje.com/) is one such service that has extensive experience helping authors meet PLOS guidelines and can provide language editing, translation, manuscript formatting, and figure formatting to ensure your manuscript meets our submission guidelines. Please note that having the manuscript copyedited by AJE or any other editing services does not guarantee selection for peer review or acceptance for publication.  Upon resubmission, please provide the following: The name of the colleague or the details of the professional service that edited your manuscript A copy of your manuscript showing your changes by either highlighting them or using track changes (uploaded as a *supporting information* file) A clean copy of the edited manuscript (uploaded as the new *manuscript* file)” 4. One of the noted authors is a group or consortium [DATs-PD GETM Spanish Registry STUDY GROUP]. In addition to naming the author group, please list the individual authors and affiliations within this group in the acknowledgments section of your manuscript. Please also indicate clearly a lead author for this group along with a contact email address. 5. Your ethics statement should only appear in the Methods section of your manuscript. If your ethics statement is written in any section besides the Methods, please move it to the Methods section and delete it from any other section. Please ensure that your ethics statement is included in your manuscript, as the ethics statement entered into the online submission form will not be published alongside your manuscript. 

Additional Editor Comments:

The authors have provided an interesting summary on device-assisted therapies in Parkinson's disease but, as the reviewers have noted, a clearer abstract followed by a balanced discussion is required. For example, demonstrating benefits and advantages of each device considered in the form of table.

Reviewers' comments:

Reviewer's Responses to Questions

**Comments to the Author**

1. Does the manuscript provide a valid rationale for the proposed study, with clearly identified and justified research questions?

Reviewer #1: Yes

Reviewer #2: Yes

2. Is the protocol technically sound and planned in a manner that will lead to a meaningful outcome and allow testing the stated hypotheses?

Reviewer #1: Yes

Reviewer #2: Yes

3. Is the methodology feasible and described in sufficient detail to allow the work to be replicable?

Reviewer #1: No

Reviewer #2: Yes

4. Have the authors described where all data underlying the findings will be made available when the study is complete?

Reviewer #1: No

Reviewer #2: Yes

5. Is the manuscript presented in an intelligible fashion and written in standard English?

Reviewer #1: Yes

Reviewer #2: No

6. Review Comments to the Author

You may also provide optional suggestions and comments to authors that they might find helpful in planning their study.

Reviewer #1: In this paper the authors present the protocol for a register for patients with PD on DAT. This is very important as there is a clear lack of comparative, longitudinal studies between different treatment options of adavnced PD.

The forst objective of the registry would be to:

1) to analyze the sociodemographic and clinical characteristics of PwP

treated with a DAT, comparing the different treatments (DBS vs subcutaneous vs enteral

treatment). It is not clear what subcutaneous (subcutavneous apomorphine pump and foslevodopa/foscarbidopa pump) and eneteral (LCIG, LECIG) DAT options would be taken into account. This needs to be sepcified.

There is an important objective missing - that is to define predictors of a favourable response to different DATs.

Could the authors forsee how long the study will last for?

"The first follow-up visit will be carried out at 6

months +/- 3 months (V3_6M) and then at 1 year +/-3 months and subsequently annually +/- 3

months: 1 year (V3_12M); 2 years (V3_24M); 3 years (V3_36M); 4 years (V3_48M); 5 years

(V3_60M); etc." ETc is not good enough. Please, specifiy - indefenete, or for at least 10 years, or up to 20 years depending on funding...

Also, Table 1 is confusing - at present, it seems there are going to be three visits only!

Figure 1. Please change 31/Dic/2033 to 31/Dec/2033 in

The ABtract needs to be imporved - it does not reflect the exat content of the paper. For example, lines 58-60 in the Abstract - It is not clear of wheterer the follow-up would last for 2, 3 years, or longer as explained later on in the paper.

Please correct typos - for example Abril to April (line 9, page 1) in the Abstract.

Instead of People with Parkinson's disease (PwP) might be Subject(s) with Parkinson's disease (SPD). Xonsider change it.

Reviewer #2: The present protocol has been drafted to inform about clinical reasoning, management, and possible beneficial as well as side effects of newly-introduced device-aided therapies, such as

levodopa-entacapone-carbidopa intestinal gel infusion (LECIG) and subcutaneous

infusion of foslevodopa/foscarbidopa (fLD/fCD).

The protocol looks good and I have some minor suggestions:

- I would avoid using the term "DAT" as it is misleading (too similar to the DaT SPECT Scan). I would rather use "DT" or avoid the abbreviated form.

- In the introduction, I would avoid using the term "parkinson plus syndrome" as it is obsolete. I would rather use "atypical parkinsonism" instead.

- In the Methods, Line 166, I would specify at point 1: "Clinical diagnosis of PD according to the MDS criteria". Please, add a citation here.

- Assessments: Are the authors planning to collect yearly MoCA and/or any cognitive screening tests? I believe this is highly informative. Also, authors use both the terms UPDRS and MDS-UPDRS. I hope they are going to use the MDS-UPDRS scale. Please, clarify. Finally, "MNCD" and "EUROHIS-QOL" should be spelled out. Please, spell out LEDD and explain the methods of calculation (e.g., Tomlinson 2010 or more recent amendments).

- English language should be carefully revised as there are many typos.

7. PLOS authors have the option to publish the peer review history of their article (what does this mean? ). If published, this will include your full peer review and any attached files.

**Do you want your identity to be public for this peer review?** For information about this choice, including consent withdrawal, please see our Privacy Policy .

Reviewer #1: No

Reviewer #2: No

---

## [Author Response · Author response to Decision Letter 1]

28 Jan 2025

Reviewer #1: In this paper the authors present the protocol for a register for patients with PD on DAT. This is very important as there is a clear lack of comparative, longitudinal studies between different treatment options of adavnced PD.

The forst objective of the registry would be to:

1) to analyze the sociodemographic and clinical characteristics of PwP

treated with a DAT, comparing the different treatments (DBS vs subcutaneous vs enteral

treatment). It is not clear what subcutaneous (subcutavneous apomorphine pump and foslevodopa/foscarbidopa pump) and eneteral (LCIG, LECIG) DAT options would be taken into account. This needs to be sepcified.

AUTHORS – Thank you very much for your comment. That is indeed the first objective into the specific objectives: “1) to analyze the sociodemographic and clinical characteristics of PwP treated with a DAT, comparing the different treatments (DBS vs subcutaneous vs enteral treatment)”. The reason for including all therapies has been justified in the introduction.

There is an important objective missing - that is to define predictors of a favourable response to different DATs.

AUTHORS – Thank you very much for your comment. We agree with you. This objective has been added. Many thanks again.

Could the authors forsee how long the study will last for?

"The first follow-up visit will be carried out at 6

months +/- 3 months (V3_6M) and then at 1 year +/-3 months and subsequently annually +/- 3

months: 1 year (V3_12M); 2 years (V3_24M); 3 years (V3_36M); 4 years (V3_48M); 5 years

(V3_60M); etc." ETc is not good enough. Please, specifiy - indefenete, or for at least 10 years, or up to 20 years depending on funding...

AUTHORS – Thank you very much for your comment. This aspect is explained in the protocol: “The period of inclusion of patients in the registry will be a minimum of 5 years, extendable to 10 years according to the approved protocol. New PwP will be included over a 10-year period, while each participant's included will be follow-up during this time. The data collection period ranges from January 1, 2024 to December 31, 2033”. As you suggest, we change the sentence and all detail data about the visits: “The first follow-up visit will be carried out at 6 months +/- 3 months (V3_6M) and then at 1 year +/-3 months and subsequently annually +/- 3 months: 1 year (V3_12M); 2 years (V3_24M); 3 years (V3_36M); 4 years (V3_48M); 5 years (V3_60M); 6 years (V3_72M); 7 years (V3_84M); 8 years (V3_96M); 9 years (V3_108M); 10 years (V3_120M) (Figure 1)”.

Also, Table 1 is confusing - at present, it seems there are going to be three visits only!

AUTHORS – Thank you very much for your comment. This point has been clarified in the table. Please, see the symbol * and the explanation: “Information collected for all these visits (with a window of +/- 3 months): 1 year (V3_12M); 2 years (V3_24M); 3 years (V3_36M); 4 years (V3_48M); 5 years (V3_60M); 6 years (V3_72M); 7 years (V3_84M); 8 years (V3_96M); 9 years (V3_108M); 10 years (V3_120M)”.

Figure 1. Please change 31/Dic/2033 to 31/Dec/2033 in

AUTHORS – Thank you very much for your comment. It has been corrected.

The ABtract needs to be imporved - it does not reflect the exat content of the paper. For example, lines 58-60 in the Abstract - It is not clear of wheterer the follow-up would last for 2, 3 years, or longer as explained later on in the paper.

AUTHORS – Thank you very much for your comment. In the abstract, the methods about the project (main and specific objectives, types of visits, start and end of recruitment, etc.) are explained. Since there is a 250 word limit we have added at the end of the methods section as you indicate "for 10 years". Other sentences have been modified.

Please correct typos - for example Abril to April (line 9, page 1) in the Abstract.

Instead of People with Parkinson's disease (PwP) might be Subject(s) with Parkinson's disease (SPD). Xonsider change it.

AUTHORS – Thank you very much for your comment. Typos have been corrected. We appreciate your suggestion but if you don´t mind we prefer to maintain the term PwP because it has been extensively used in the literature compared with the other term (https://pubmed.ncbi.nlm.nih.gov/?term=PwP+people+with+parkinson&sort=date).

Reviewer #2: The present protocol has been drafted to inform about clinical reasoning, management, and possible beneficial as well as side effects of newly-introduced device-aided therapies, such as levodopa-entacapone-carbidopa intestinal gel infusion (LECIG) and subcutaneous infusion of foslevodopa/foscarbidopa (fLD/fCD).

The protocol looks good and I have some minor suggestions:

- I would avoid using the term "DAT" as it is misleading (too similar to the DaT SPECT Scan). I would rather use "DT" or avoid the abbreviated form.

AUTHORS – Thank you for your comment. I understand your suggestion. We have decided to use DAT because is the abbreviation used in the literature of device-aided therapy. In all the recent articles about this topic, DAT is always used (https://pubmed.ncbi.nlm.nih.gov/?term=DATs+device-aided+therapies). This is why we understand the importance of using the term DAT to ensure consistency in the literature and facilitate the dissemination of the work.

- In the introduction, I would avoid using the term "parkinson plus syndrome" as it is obsolete. I would rather use "atypical parkinsonism" instead.

AUTHORS – Thank you for your comment. The term has been changed.

- In the Methods, Line 166, I would specify at point 1: "Clinical diagnosis of PD according to the MDS criteria". Please, add a citation here.

AUTHORS – Thank you for your comment. This is the reference 3. It has been added: “1) diagnosis of PD according to the MDS criteria [3];”.

- Assessments: Are the authors planning to collect yearly MoCA and/or any cognitive screening tests? I believe this is highly informative. Also, authors use both the terms UPDRS and MDS-UPDRS. I hope they are going to use the MDS-UPDRS scale. Please, clarify. Finally, "MNCD" and "EUROHIS-QOL" should be spelled out. Please, spell out LEDD and explain the methods of calculation (e.g., Tomlinson 2010 or more recent amendments).

AUTHORS – Thank you for your comment. In patients with cognitive impairment, the MMSE, MOCA or PD-CRS may be used at the discretion of the neurologist in daily clinical practice, but due to problems with sufficient time to collect the information, it will not be performed in a regulated manner in all patients. We have added this information: “non-motor symptoms including cognition; the Mini-Mental State Examination (MMSE) [16], Montreal Cognitive Assessment (MoCA) [17] or Parkinson´s Disease Cognitive Rating Scale (PD-CRS) [18] may be used at the discretion of the neurologist in daily clinical practice)”. The protocol explains that the UPDRS-III will be used and that, in addition, in some centers that use it regularly in clinical practice, information from the MDS-UPDRS-III could also be collected. In line 219, the term LEDD appears spelt out and there is a reference about how to calculate it: “levodopa equivalent daily dose (LEDD) [15]”. We clarify it in the table. In addition, MNCD" and "EUROHIS-QOL” have been spelt out.

- English language should be carefully revised as there are many typos.

AUTHORS – Thank you very much for your comments. English has been reviewed and typos have been corrected. Specifically, it has been reviewed by a worker of Scholar Memory Academic Writing Consultant. We greatly appreciate his help and include it in the acknowledgements.

---

## [Editor Report · Decision Letter 1]

10 Feb 2025

Device-aided therapies (DATs) in Parkinson´s disease (PD). The DATs-PD GETM Spanish Registry.

PONE-D-24-54866R1

Dear Dr. Garcia,

We’re pleased to inform you that your manuscript has been judged scientifically suitable for publication and will be formally accepted for publication once it meets all outstanding technical requirements.

Kind regards,

Barry Kweh

Academic Editor

PLOS ONE

Additional Editor Comments (optional):

The authors have revised their manuscript by improving the methodological rigour, clarifying terminology such as DAT and provided adequate statistical justification.
---

## [Editor Report · Acceptance letter]

PONE-D-24-54866R1

PLOS ONE

Dear Dr. Santos Garcia,

I'm pleased to inform you that your manuscript has been deemed suitable for publication in PLOS ONE. Congratulations! Your manuscript is now being handed over to our production team.

Kind regards,

on behalf of

Dr. Barry Kweh

Academic Editor

PLOS ONE